# Temporal Alignment Prediction for Supervised Representation Learning and Few-Shot Sequence Classification

**Bing Su & Ji-Rong Wen**[*]
Beijing Key Laboratory of Big Data Management and Analysis Methods
Gaoling School of Artificial Intelligence, Renmin University of China, Beijing 100872, China
`subingats@gmail.com; jrwen@ruc.edu.cn`

## Abstract

Explainable distances for sequence data depend on temporal alignment to tackle sequences with different lengths and local variances. Most sequence alignment methods infer the optimal alignment by solving an optimization problem under pre-defined feasible alignment constraints, which not only is time-consuming, but also makes end-to-end sequence learning intractable. In this paper, we propose a learnable sequence distance called Temporal Alignment Prediction (TAP). TAP employs a lightweight convolutional neural network to directly predict the optimal alignment between two sequences, so that only straightforward calculations are required and no optimization is involved in inference. TAP can be applied in different distance-based machine learning tasks. For supervised sequence representation learning, we show that TAP trained with various metric learning losses achieves completive performances with much faster inference speed. For few-shot action classification, we apply TAP as the distance measure in the metric learning-based episode-training paradigm. This simple strategy achieves comparable results with state-of-the-art few-shot action recognition methods.

## 1 Introduction

Distance between sequences plays a crucial role in sequence classification (Sakoe & Chiba, 1978), retrieval (Su et al., 2019), clustering (García-García et al., 2008), etc. Measuring distance between sequences is difficult, since different sequences may have different sampling rates, execution speeds, local distortions, initial states, and elastic warpage. To tackle such temporal variances, existing sequence distances either encode each sequence into a feature vector invariant to temporal variances (Abid & Zou, 2018; Lohit et al., 2019) or employ alignment for temporal correspondence calibration (Sakoe & Chiba, 1978; Su & Hua, 2019). Typical feature-based methods use recurrent neural networks (RNNs) (Ramachandran et al., 2017) to encode sequences and measure the Euclidean distance between corresponding features. The feature of a sequence is fixed when compared with any other sequence. Although such methods are naturally learnable and only perform forward calculations in inference, they require large amounts of sequences to train complex RNNs. Moreover, how the learned features handle temporal variances and what kinds of variances can be handled are not clear.

Alignment-based methods determine different optimal alignments for different sequence pairs. This is more intuitive and flexible because temporal variances may be different when comparing different sequences. The inferred alignments clearly indicate how and where the two sequences differ in temporal steps. Most alignment methods solve an optimization problem under pre-defined feasible constraints to infer the optimal alignment. E.g., DTW (Sakoe & Chiba, 1978) requires dynamic programming and OPW (Su & Hua, 2019) employs fixed-point iterations. Such optimizations are often time-consuming and cannot fully utilize GPU.

Moreover, since inferring the alignment is itself an optimization problem and has its own objective, sequence distance-based end-to-end learning using other objectives becomes intractable. For

---

[*]Corresponding author: Ji-Rong Wen.

instance, learning discriminative temporal representations for elements in sequences often adopt the objective that sequences of different classes are better separated w.r.t. a sequence distance (Mei et al., 2014; Su & Wu, 2020b). Gradients of this overall objective are difficult to pass through alignments since they are latent variables determined by another optimization problem.

In this paper, we propose a learnable alignment-based sequence distance, namely *Temporal Alignment Prediction (TAP)*. TAP simulates the optimization-based inference from the input sequence pair to the optimal alignment as a function. The function is modeled by a lightweight convolutional neural network to directly predict the optimal alignment. TAP can be applied in different distance-based machine learning tasks under different settings. As instances, we show the applications of TAP in supervised representation learning and few-shot learning for sequence data. For supervised learning, we employ metric learning losses by using TAP as the distance measure to learn the frame-wise temporal representations. For few-shot learning, we adopt TAP as the distance measure in the metric learning based paradigm to compare the query and support sequence samples. In both cases, owing to the straightforward structure of TAP, the alignment prediction network and the feature extraction or transformation network can be jointly trained in an end-to-end manner, resulting in principled yet straightforward supervised learning and few-shot learning solutions for sequences. We further show the application of TAP in self-supervised alignment learning in the appendix.

The main contributions of this paper are three-fold. 1. We propose TAP, a learnable and fast sequence distance that only requires straightforward calculations in inference. 2. We show that TAP conveniently enables end-to-end supervised sequence representation learning with different losses. 3. We adopt TAP to dynamically align the support and query sequences for few-shot action recognition, so that alignments and temporal representations can be jointly learned with the episode-training strategy. Experiments on seven datasets demonstrate the effectiveness of TAP for both supervised learning and few-shot learning.

## 2 RELATED WORK

**Alignment-based sequence distance.** Various variations of DTW have been proposed to speed up the inference (Salvador & Chan, 2007; Al-Naymat et al., 2009), or adapt to additional or modified constraints (Ratanamahatana & Keogh, 2004), or tackle sequences from different modalities (Zhou & Torre, 2009; Zhou & De la Torre, 2016; Trigeorgis et al., 2016; 2017; Cohen et al., 2021). They rely on non-differential dynamic programming for inference. Soft-DTW (Cuturi & Blondel, 2017) optimizes a differentiable loss by using the soft-minimum cost of all feasible alignments as the objective. TCOT (Su et al., 2019) and OPW (Su & Hua, 2019) view the alignment problem as the optimal transport problem with temporal regularizations. These methods solve an optimization problem to infer the optimal alignment. In contrast, our method directly predicts the alignment using a neural network, does not have its own objective, and only involves differentiable calculations.

**Data-driven alignment methods.** In Abid & Zou (2018), Autowarp finds the warping distance that best mimics the distance between the features encoded by LSTM, which is difficult to be integrated into an end-to-end framework because the autoencoder needs to be pre-trained and a betaCV-based optimization is required. It is shown in Tallec & Ollivier (2018) that RNN can learn to warp sequences. In Lohit et al. (2019); Weber et al. (2019), a fixed warping function is predicted for a given sequence when comparing with different sequences. This may be viewed as warping all sequences w.r.t. a fixed timing scale, which may bias to the training sequences. In contrast, since the relative local variations and correspondences between different sequences are different, our method predicts different alignments for different sequence pairs.

**Supervised representation learning for sequences.** Generally, since alignment-based distances need to solve their own optimization problems to infer alignments in the latent space, the gradients of the supervised metric learning loss cannot be directly propagated. Existing methods exploit either an approximate proxy loss (Su et al., 2018; Lajugie et al., 2014), or a time-consuming optimization method to iteratively infer alignments (Mei et al., 2014), or both (Su & Wu, 2020a;b). In this paper, we show that existing deep metric learning methods can be readily applied to the proposed TAP distance in an end-to-end manner.

**Few-shot action recognition.** Most existing few-shot action recognition methods follow the metric learning based paradigm. In Ben-Ari et al. (2021) and Tan & Yang (2019), TAEN and FAN en-

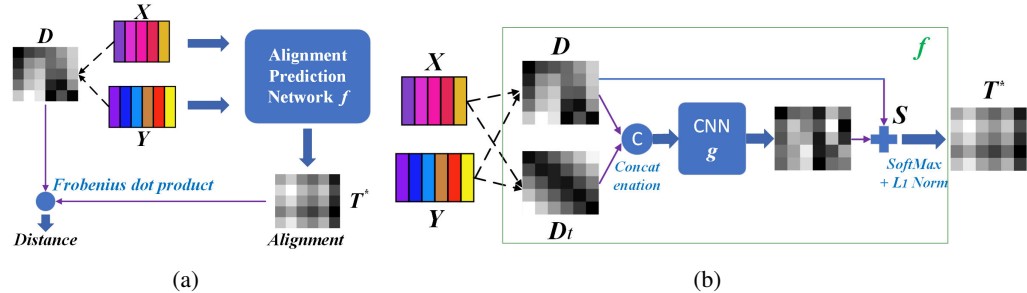

Figure 1: (a) The TAP framework. (b) The alignment prediction network.

code each action into a representation with a fixed dimension and apply vector-wise metrics. Recent works identify the importance of temporal alignment for tackling non-linear temporal variations. Various alignment-based distances are proposed to compare sequence pairs, such as the temporal attentive relation network (TARN) (Bishay et al., 2019), a variant of DTW used in OTAM (Cao et al., 2020), the permutation-invariant spatial and temporal attention reweighted distance in ARN (Zhang et al., 2020), the temporal CrossTransformer for comparing all possible subsequences in TRX (Perrett et al., 2021), and the two-stage temporal alignment network (TTAN) (Li et al., 2021). Following this spirit, we apply the proposed TAP to perform temporal alignment and measure the distance between support and query sequences without any complicated attention mechanism, yielding a simple yet effective few-shot sequence learning solution.

## 3 METHODOLOGY

### 3.1 TEMPORAL ALIGNMENT PREDICTION

We first revisit the unified formulation of sequence distance in Su & Wu (2019). Let $\boldsymbol{X} = [\boldsymbol{x}_1, \cdots, \boldsymbol{x}_{L_X}] \in \mathbb{R}^{d \times L_X}$ and $\boldsymbol{Y} = [\boldsymbol{y}_1, \cdots, \boldsymbol{y}_{L_Y}] \in \mathbb{R}^{d \times L_Y}$ be two sequences with lengths of $L_X$ and $L_Y$, respectively. The elements $\boldsymbol{x}_i, i = 1, \cdots, L_X$ of $\boldsymbol{X}$ and $\boldsymbol{y}_j, j = 1, \cdots, L_Y$ of $\boldsymbol{Y}$ lie in a $d$-dimensional feature space $\mathbb{R}^d$. Many alignment-based distances between $\boldsymbol{X}$ and $\boldsymbol{Y}$ can be unified as follows:

$$d(\boldsymbol{X}, \boldsymbol{Y}) = \langle \boldsymbol{T}^*, \boldsymbol{D} \rangle, \tag{1}$$

$$\boldsymbol{T}^* = \underset{\boldsymbol{T} \in \boldsymbol{\Phi}}{arg\min} \langle \boldsymbol{T}, \boldsymbol{D} \rangle + \mathscr{R}(\boldsymbol{T}), \tag{2}$$

where $\langle \boldsymbol{T}, \boldsymbol{D} \rangle = tr(\boldsymbol{T}^T \boldsymbol{D})$ is the Frobenius dot product. $\boldsymbol{D} := [e(\boldsymbol{x}_i, \boldsymbol{y}_j)]_{ij} \in \mathbb{R}^{L_X \times L_Y}$ is the matrix of pairwise distances between elements in $\boldsymbol{X}$ and $\boldsymbol{Y}$, $e(\boldsymbol{x}_i, \boldsymbol{y}_j)$ is a vector-wise distance between two elements $\boldsymbol{x}_i$ and $\boldsymbol{y}_j$, which we use Euclidean distance in this paper. $\boldsymbol{T}$ is an alignment matrix whose element $\boldsymbol{T}_{ij}$ indicates whether or how likely $\boldsymbol{x}_i$ and $\boldsymbol{y}_j$ are aligned. $\boldsymbol{\Phi}$ is the set of all feasible $\boldsymbol{T}$ with some constraints, which is a subset of $\mathbb{R}^{L_X \times L_Y}$. $\mathscr{R}(\boldsymbol{T})$ is a regularization term on $\boldsymbol{T}$. $\boldsymbol{T}^*$ is the optimal alignment which is the solution of the optimization problem in Eq. (2).

Different sequence distances impose different constraints on the feasible set, have different regularization terms, and use different optimization methods for inference. For instance, DTW optimizes Eq. (2) with $\mathscr{R}(\boldsymbol{T}) = 0$ and the boundary, continuity, and monotonicity constraints via dynamic programming, while OPW optimizes Eq. (2) with two temporal regularization terms and the coupling constraints via the Sinkhorn's matrix scaling algorithm. Solving $\boldsymbol{T}^*$ by the optimization Eq. (2) not only requires a long inference time, but also makes it difficult to apply a loss on the sequence distance Eq. (1) for learning element representations, because $\boldsymbol{T}^*$ is a latent variable that needs to be inferred and its gradient cannot be calculated.

To avoid solving the optimization problem, we propose a feedforward framework for measuring the distances between sequences, namely *Temporal Alignment Prediction (TAP)*. Fig. 1(a) illustrates the TAP framework. For two sequences $\boldsymbol{X} = [\boldsymbol{x}_1, \cdots, \boldsymbol{x}_{L_X}]$ and $\boldsymbol{Y} = [\boldsymbol{y}_1, \cdots, \boldsymbol{y}_{L_Y}]$, their TAP distance also has the form of Eq. (1), i.e., the Frobenius dot product of $\boldsymbol{D}$ and $\boldsymbol{T}^*$. Different from other sequence distances which infer the alignment with predefined objectives and constraints, TAP uses an alignment prediction neural network $f$ to directly predict the optimal $\boldsymbol{T}^* = f(\boldsymbol{X}, \boldsymbol{Y})$ by taking the two sequences as inputs and learns $f$ from data.

The alignment prediction network $f$ can be instantiated by different architectures. In this paper, we propose a lightweight convolutional neural network architecture for its simplicity and fast inference speed. Specifically, we reuse the spatial ground matrix $\boldsymbol{D}$ to measure all pairwise distances between elements in $\boldsymbol{X}$ and $\boldsymbol{Y}$. The relative positions of $\hat{\boldsymbol{x}}_i$ and $\hat{\boldsymbol{y}}_j$ are $i/L_X$ and $j/L_Y$, respectively. To incorporate the temporal dissimilarities, TAP further calculates all pairwise Euclidean distances between the relative positions between $\boldsymbol{x}_i$ and $\boldsymbol{y}_j$ into a matrix $\boldsymbol{D}_t := [e(i/L_X, j/L_Y)]_{ij} \in \mathbb{R}^{L_X \times L_Y}$.

Then, we concatenate $\boldsymbol{D}$ and $\boldsymbol{D}_t$ along the channel dimension to form $\boldsymbol{D}_s$ with a size of $L_x \times L_y \times 2$. We use a CNN $g$ to predict the final alignment matrix from $\boldsymbol{D}_s$. The CNN uses three convolutional layers each followed by ReLu. The kernel size in the three layers is 5, 5, and 3, respectively, the stride is fixed to 1, and the padding is set to 2, 2, and 1, respectively, to keep the spatial size. The numbers of kernels in the three layers are set to 30, 30, and 1, respectively. The learned kernels are expected to capture the local alignment patterns. The output of $f$ is augmented by a residual connection with $\boldsymbol{D}$, resulting in the similarity matrix $\boldsymbol{S} := -(\boldsymbol{D} + g(\boldsymbol{D}_s)) \in \mathbb{R}^{L_X \times L_Y}$.

The attentions of elements in $\boldsymbol{Y}$ on $\boldsymbol{x}_i$ are obtained by performing Softmax on the $i$-th row of $\boldsymbol{S}$. The attentions on all elements in $\boldsymbol{X}$ form an attention matrix $\boldsymbol{A}$, which can be obtained by performing Softmax along the 2nd dimension of $\boldsymbol{S}$. To generate the predicted alignment $\boldsymbol{T}^*$, TAP finally performs a global $L_1$ normalization on $\boldsymbol{A}$:

$$\boldsymbol{A} = [\frac{exp(\boldsymbol{S}_{ij})}{\sum_{k=1}^{L_Y} exp(\boldsymbol{S}_{ik})}]_{ij} \in \mathbb{R}^{L_X \times L_Y}; \qquad \boldsymbol{T}^* = [\frac{\boldsymbol{A}_{ij}}{\sum_{i=1}^{L_X} \sum_{j=1}^{L_Y} \boldsymbol{A}_{ij}}]_{ij} \in \mathbb{R}^{L_X \times L_Y}. \qquad (3)$$

$\boldsymbol{T}^*_{ij}$ indicates the probability of aligning $\boldsymbol{x}_i$ and $\boldsymbol{y}_j$. The TAP distance between $\boldsymbol{X}$ and $\boldsymbol{Y}$ is calculated as in Eq. (1). The prediction network $f$ of TAP is lightweight since it only contains a few convolutional kernels of $g$ as parameters.

**Limitations.** 1. The family of alignments for TAP is $\Phi := \{\boldsymbol{T} \in \mathbb{R}_+^{L_x \times L_y} | \boldsymbol{T} \boldsymbol{1}_{L_y} = \boldsymbol{1}_{L_x}/L_x\}$, where $\boldsymbol{1}_L$ is a $L$-dimensional vector with all 1 elements. Since strict order-preserving is not guaranteed in $\Phi$, the performance of TAP may be limited when data are strictly ordered. 2. $\boldsymbol{T}^T \boldsymbol{1}_{L_x} = \boldsymbol{1}_{L_y}/L_y$ does not necessarily hold, hence TAP is asymmetric and is not a real metric. These limitations make TAP more flexible in turn. 1. Without strict order-preserving constraint, TAP can tackle local temporal reorders and generalize to non-sequential (e.g. spatial, cross-modal, etc) correspondences. 2. Asymmetric alignment distinguishes the source sequence $\boldsymbol{X}$ and the target sequence $\boldsymbol{Y}$, where all elements in $\boldsymbol{X}$ must be transported with the same mass when aligning to different target sequences. To perform classification or retrieval, we can always set the test or query sequence as the source, which serves as a standard template to be aligned. The symmetric distance can be obtained by averaging $d(\boldsymbol{X}, \boldsymbol{Y})$ and $d(\boldsymbol{Y}, \boldsymbol{X})$.

## 3.2 SUPERVISED REPRESENTATION LEARNING WITH TAP

The performance of conventional alignment-based distances and TAP highly depends on $\boldsymbol{D}$, where $\boldsymbol{D}$ depends on the representation space or the ground metric for elements in sequences. Therefore, learning more discriminative representations or ground metrics is of great interest. We transform element-wise features $\boldsymbol{x}_i$ of $\boldsymbol{X}$ into a new space by a transformation network $h$. We denote the sequence of transformed features $h(\boldsymbol{x}_i)$ by $h(\boldsymbol{X})$. When labeled training sequences are available, the objective is often to make the distance better separate sequences from different classes. However, such objectives are difficult to optimize for conventional alignment-based distances because they solve another optimization problem to obtain the alignments, as presented in Appendix A.2.

In contrast, TAP enables supervised learning of temporal representations without seeking surrogates objectives. Given a batch of $B$ training sequences $\boldsymbol{X}^n, n = 1, \cdots, B$ with corresponding labels $y^n, n = 1, \cdots, B$, for each sequence $\boldsymbol{X}^n$, we generate its augmentation $\tilde{\boldsymbol{X}}^n$, whose class label is the same as $\boldsymbol{X}^n$, and calculate its TAP distances to all other $B$ sequences including $\tilde{\boldsymbol{X}}^n$ after transformation by $h$. We obtain its largest TAP distance to $h(\boldsymbol{X}^{n'})$ of the same class as $d_l^n = \max_{n':y^{n'}=y^n} d(h(\boldsymbol{X}^n), h(\boldsymbol{X}^{n'}); f)$ and its smallest TAP distance to $h(\boldsymbol{X}^{n'})$ of different classes as $d_s^n = \min_{n':y^{n'}\neq y^n} d(h(\boldsymbol{X}^n), h(\boldsymbol{X}^{n'}); f)$.

Any metric learning losses can be used to train TAP. We employ three metric learning losses to learn $h$ and $f$ jointly, including the triplet loss with hard-mining (HM) (Hermans et al., 2017), the lifted

structure (Lifted) loss (Song et al., 2016), and binomial deviance loss (Binomial) (Yi et al., 2014). For the triplet loss, we minimize the TAP distances to sequences of the same class that are larger than $d_s^n - m$, and maximize those to different classes that are smaller than $d_l^n + m$, where $m$ is a margin and set to $0.1$. The objective to be minimized over $f$ and $h$ is formulated as follows:

$$\min_{f,h} \sum_{n=1}^{B} \Big( \sum_{n':y^{n'}=y^n} \max(d(h(\boldsymbol{X}^n), h(\boldsymbol{X}^{n'}); f) - d_s^n + m, 0) \\ + \sum_{n':y^{n'} \neq y^n} \max(d_l^n + m - d(h(\boldsymbol{X}^n), h(\boldsymbol{X}^{n'}); f), 0) \Big) \tag{4}$$

For the lifted structure loss, we minimize the following objective to utilize the TAP distances among all the positive and negative pairs:

$$\min_{f,h} \sum_{n=1}^{B} \Big( \tfrac{2}{\beta} \log \big( \sum_{n':y^{n'}=y^n} e^{\beta d(h(\boldsymbol{X}^n), h(\boldsymbol{X}^{n'}); f)} \big) + \tfrac{2}{\alpha} \log \big( \sum_{n':y^{n'} \neq y^n} e^{-\alpha d(h(\boldsymbol{X}^n), h(\boldsymbol{X}^{n'}); f)} \big) \Big) . \tag{5}$$

For the binomial deviance loss, for the $n$-th sequence, we first perform hard mining to select its hard positive and negative samples within the batch:

$$\mathscr{P}^n = \{ n' | y^{n'} = y^n; d(h(\boldsymbol{X}^n), h(\boldsymbol{X}^{n'}); f) > d_s^n - m \} \\ \mathscr{N}^n = \{ n' | y^{n'} \neq y^n; d(h(\boldsymbol{X}^n), h(\boldsymbol{X}^{n'}); f) < d_l^n + m \} \tag{6}$$

We then minimize the following objective:

$$\min_{f,h} \sum_{n=1}^{B} \Big( \tfrac{2}{\beta |\mathscr{P}^n|} \sum_{n' \in \mathscr{P}^n} \log(1 + e^{\beta(d(h(\boldsymbol{X}^n), h(\boldsymbol{X}^{n'}); f) - 0.5)}) \\ + \tfrac{2}{\alpha |\mathscr{N}^n|} \sum_{n' \in \mathscr{N}^n} \log(1 + e^{-\alpha(d(h(\boldsymbol{X}^n), h(\boldsymbol{X}^{n'}); f) - 0.5)}) \Big) . \tag{7}$$

where $|\mathscr{P}^n|$ and $|\mathscr{N}^n|$ denote the number of mined positive and negative sequences, respectively. In all losses, $\alpha$ and $\beta$ are hyper-parameters controlling the weights of positive and negative pairs, respectively. For all the losses, we also incorporate the self-supervised loss of minimizing the TAP distance between each sequence and its augmentation, as presented in the appendix, where the augmentation is obtained by random blurring and random merge. For any supervised loss, the feature transformation network $h$ and the alignment prediction network $f$ in TAP are learned jointly by back-propagation in an end-to-end manner.

### 3.3 FEW-SHOT ACTION RECOGNITION WITH TAP

In this subsection, we demonstrate the application of TAP in few-shot action recognition. For a $N$-way $K$-shot classification task, there are $N$ action classes and each class has $K$ support videos. The task aims to classify each query video into one of the $N$ classes. We follow the episode training and evaluation. The video dataset is divided into a training set, a validation set, and a test set. Classes in the three sets are disjoint. In the training phase, we randomly construct one $N$-way $K$-shot task in each episode to train the model. The best model is selected according to the performance on tasks constructed from the validation set and evaluated on tasks constructed from the test set.

We adopt the metric learning strategy with the TAP distance. For each $N$-way $K$-shot classification task, the support videos are denoted by $\boldsymbol{V}_k^n, n = 1, \cdots, N; k = 1, \cdots, K$ and the query videos are denoted by $\boldsymbol{V}_q, q = 1, \cdots, Q$, where $Q$ is the number of query videos. We follow Cao et al. (2020); Li et al. (2021) to sample a fixed number $L$ of frames per video. We use a feature embedding backbone $h$ to represent each frame $\boldsymbol{v}_i$ by a $d$-dimensional feature vector $\boldsymbol{x}_i = h(\boldsymbol{v}_i)$, thus each video $\boldsymbol{V}$ is represented by a feature sequence $\boldsymbol{X} = h(\boldsymbol{V}) = [\boldsymbol{x}_1, \cdots, \boldsymbol{x}_L] \in \mathbb{R}^{d \times L}$. A self-attention (SA) module can be performed to enhance $\boldsymbol{X} = SA(\boldsymbol{X})$ as presented in Appendix A.3.

To classify a query video $\boldsymbol{V}_q$, we calculate the TAP distances between $\boldsymbol{X}_q$ and all the support sequences. Since TAP is non-symmetric, we add $d(\boldsymbol{X}_q, \boldsymbol{X}_k^n; f)$ and $d(\boldsymbol{X}_k^n, \boldsymbol{X}_q; f)$ as the dissimilarity between $\boldsymbol{X}_q$ and the $k$-th support sequence of the $n$-th class $\boldsymbol{X}_k^n$. The similarity between $\boldsymbol{V}_q$ and the $n$-th class is then obtained as:

$$s_q^n = \sum_{k=1}^{K} -d(\boldsymbol{X}_q, \boldsymbol{X}_k^n; f) - d(\boldsymbol{X}_k^n, \boldsymbol{X}_q; f) \tag{8}$$

$\boldsymbol{X}_q$ is classified to the class with the largest similarity.

In the training phase, we jointly learn the backbone $h$ and the prediction network $f$ of TAP with the cross-entropy loss. The loss over a single episode is formulated as follows:

$$\mathcal{L} = -\sum_{q=1}^{Q} \log \frac{exp(s_q^{y_q})}{\sum_{n=1}^{N} exp(s_q^{n})}. \tag{9}$$

where $y_q$ is the ground-truth class label of the query video $\boldsymbol{V}_q$. If SA is applied to query and support sequences before performing alignment by TAP, the method is denoted by SA-TAP.

## 4 EXPERIMENTS

### 4.1 DATASETS

For supervised representation learning, we evaluate TAP on four sequence datasets. **MSR Action3D** (Li et al., 2010; Wang et al., 2012) contains 557 skeleton sequences from 20 action classes. We follow the splitting of the dataset in Wang et al. (2012); Wang & Wu (2013). We use the 192-dimensional element-wise features in Wang et al. (2012); Wang & Wu (2013). **MSR Daily Activity3D** (Wang et al., 2012) contains 320 skeleton sequences from 16 activity classes. We also follow the splitting of the dataset in Wang et al. (2012); Wang & Wu (2013) and use the 390-dimensional element-wise features in Wang et al. (2012); Wang & Wu (2013). The **"Spoken Arabic Digits (SAD)" dataset** from the UCI Machine Learning Repository (Bache & Lichman, 2013) contains 8,800 sequences of 13-dimensional mel-frequency cepstrum coefficients from 10 classes. The dataset has 6,600 training sequences and 2,200 testing sequences. **ChaLearn** (Escalera et al., 2013b;a) contains 955 Italian gesture sequences from 20 classes. The dataset is split into training, validation, and test sets. Following Fernando et al. (2015), we conduct experiments on the segmented sequences where each sequence contains one gesture. We use the 100-dimensional element-wise features in Fernando et al. (2015). On all datasets, the lengths of different sequences are different.

For few-shot action recognition, we evaluate TAP on three video datasets. **UCF101** (Soomro et al., 2012) contains 13,320 videos of 101 classes. We follow the protocol used in Zhang et al. (2020) to split the dataset, where the training, validation, and test sets contain 70, 10, and 21 classes with 9154, 1421, and 2745 videos, respectively. **HMDB51** (Kuehne et al., 2011) consists of 6,840 videos from 51 classes. We also follow the protocol used in Zhang et al. (2020) to split the dataest, where the training, validation, and test sets contain 31, 10, and 10 classes with 4280, 1194, and 1292 videos, respectively. **Something-Something-v2 (SSv2)** (Goyal et al., 2017) contains 220,847 videos of 174 classes. We follow the protocol in Cao et al. (2020) to split the dataset, where the training, validation, and test sets contain 64, 12, and 24 classes with 77,500, 1925, and 2854 videos, respectively.

### 4.2 IMPLEMENTATION DETAILS

The structure and hyper-parameters of the alignment prediction network are fixed in TAP, as introduced in Section 3.1. For supervised representation learning, we perform two tasks on each dataset: sequence classification and sequence retrieval. For classification, we use the nearest neighbor (1-NN) classifier. For each test sequence, we calculate the distances to all training sequences w.r.t. the evaluated distance and its label is determined by the training sequence with the smallest distance. We use accuracy as the performance measure. For retrieval, we use each test sequence as a query and rank all training sequences according to their distances to the query in ascending order. We use mean average precision (MAP) as the performance measure.

Following Su & Wu (2020b), we use a three-layer perceptron as the transformation network $h$, where the two hidden layers have 1024 nodes followed by ReLU activations and the output has the same dimension with the input element-wise features. For learning the TAP distance, we use the Adam optimizer with a momentum of 0.9 and a weight decay of $1e^{-4}$ and train a maximum of 100 epochs. The learning rate is fixed to 0.0001 on all datasets. The batch size is set to 64 except for the MSR Activity3D dataset, where it is reduced to 16 due to memory limitation. We fix the hyper-parameters $\alpha$ and $\beta$ in the lifted structure loss and the binomial deviance loss to 40 and 2, respectively.

To evaluate TAP on few-shot action recognition, we perform 5-way 1-shot and 5-way 5-shot classification tasks. We increase the numbers of kernels in the middle layers of $f$ from 30 to 64. Fol-

Table 1: Comparison of supervised representation learning methods.

| Method | Action3D | | Activity3D | | SAD | | ChaLearn | |
|---|---|---|---|---|---|---|---|---|
| | MAP | 1-NN | MAP | 1-NN | MAP | 1-NN | MAP | 1-NN |
| LDMLT+DTW (Mei et al., 2014) | 0.643 | 84.98 | 0.366 | 55.00 | 0.595 | 96.50 | 0.213 | 84.37 |
| LDMLT+OPW (Mei et al., 2014) | 0.536 | 80.59 | 0.348 | 54.37 | 0.611 | 96.73 | 0.216 | 82.74 |
| SWMD+DTW (Huang et al., 2016) | 0.597 | 80.95 | 0.378 | 61.25 | 0.524 | 93.95 | 0.144 | 64.45 |
| SWMD+OPW (Huang et al., 2016) | 0.432 | 66.67 | 0.356 | 55.00 | 0.580 | 95.41 | 0.154 | 60.31 |
| RVSML+DTW (Su & Wu, 2019) | 0.593 | 82.78 | 0.422 | 62.50 | 0.602 | 96.23 | 0.338 | **87.38** |
| RVSML+OPW (Su & Wu, 2019) | 0.475 | 76.56 | 0.366 | 57.50 | 0.656 | 97.09 | 0.331 | 83.82 |
| Binomial+DTW (Yi et al., 2014) | 0.481 | 60.81 | 0.568 | 62.50 | 0.482 | 91.73 | 0.182 | 69.72 |
| Binomial+DTW (Yi et al., 2014) | 0.500 | 62.64 | 0.616 | 60.62 | 0.602 | 95.77 | 0.199 | 69.17 |
| Lifted+DTW (Song et al., 2016) | 0.440 | 67.40 | 0.484 | 62.50 | 0.464 | 89.64 | 0.130 | 66.85 |
| Lifted+OPW (Song et al., 2016) | 0.504 | 65.20 | 0.596 | 65.62 | 0.548 | 94.86 | 0.146 | 67.66 |
| HM+DTW (Hermans et al., 2017) | 0.432 | 57.51 | 0.589 | 65.62 | 0.656 | 92.68 | 0.237 | 67.28 |
| HM+OPW (Hermans et al., 2017) | 0.438 | 55.31 | 0.613 | 63.12 | 0.738 | 95.95 | 0.256 | 68.41 |
| MS+DTW (Wang et al., 2019) | 0.344 | 45.42 | 0.479 | 51.25 | 0.580 | 85.64 | 0.181 | 65.95 |
| MS+OPW (Wang et al., 2019) | 0.342 | 40.66 | 0.480 | 50.00 | 0.631 | 92.27 | 0.197 | 63.67 |
| D-R+DTW (Su & Wu, 2020b) | 0.617 | 79.49 | 0.531 | 61.88 | 0.782 | 97.32 | 0.440 | 81.15 |
| D-R+OPW (Su & Wu, 2020b) | 0.691 | 74.36 | 0.689 | 65.62 | 0.831 | 98.91 | 0.463 | 79.91 |
| HM+SoftDTW | 0.818 | 87.91 | 0.585 | 69.38 | 0.959 | **99.55** | 0.578 | 87.12 |
| Lifted+SoftDTW | 0.074 | 4.03 | 0.108 | 6.25 | 0.123 | 10.00 | 0.560 | 80.66 |
| Binomial+SoftDTW | 0.806 | **89.74** | 0.108 | 6.25 | 0.123 | 10.00 | 0.524 | 81.38 |
| **HM+TAP (Ours)** | 0.798 | 87.18 | 0.701 | **74.38** | **0.961** | 98.82 | **0.614** | 85.06 |
| **Lifted+TAP (Ours)** | 0.772 | 82.05 | **0.728** | 73.75 | 0.936 | 98.14 | 0.469 | 74.64 |
| **Binomial+TAP (Ours)** | **0.826** | 87.91 | 0.678 | 73.13 | 0.952 | 98.64 | **0.614** | 87.17 |

lowing Cao et al. (2020); Zhang et al. (2020), we uniformly sample $L = 8$ frames per video, resize each frame to $256 \times 256$, augment the frames with random horizontal flipping, perform $224 \times 224$ random crops for training and center crops for testing, and use the pretrained ResNet-50 as the backbone. We jointly tune the backbone and train TAP by the SGD optimizer with a learning rate of 0.001. Each episode contains 1 task due to the memory constraint. We adapt the toolbox [1] and all other hyper-parameters remain the same. We train a maximum of 100000 iterations on SSv2 and 35000 iterations on other datasets. We select the best iteration every 5000 iterations according to the performance on the validation set. For 1-shot task on HMDB51, the validation accuracy starts to fluctuate after 5000 iterations, so we train 5000 iterations and select every 1000 iterations. We randomly sample 10,000 tasks from the test set and report the average accuracy.

### 4.3 COMPARISON WITH SUPERVISED REPRESENTATION LEARNING METHODS

TAP can be used to learn temporal representations in a supervised manner as presented in Section 3.2. Comparisons with other supervised sequence representation learning methods on the four datasets are shown in Table 1, where we follow the setting and compare with the results reported in Su & Wu (2020b). LDMLT (Mei et al., 2014), SWMD (Huang et al., 2016), and RVSML (Su & Wu, 2019) learn linear transformations for elements in sequences, while deep metric learning methods with binomial deviance loss (Binomial) (Yi et al., 2014), lifted structured loss (Lifted) (Song et al., 2016), triplet loss with hard-mining (HM) (Hermans et al., 2017), and multi-similarity loss (MS) (Wang et al., 2019) learn non-linear temporal representations by treating elements in each sequence as independent samples of the same class. DeepRVSML (D-R) (Su & Wu, 2020b) is the deep extension of RVSML. All methods use the same transformation network. TAP generally outperforms these methods on all datasets. Especially, TAP achieves much higher MAP, which reflects that the TAP distances between sequences from the same class are generally smaller than those from different classes. This shows the effectiveness of the end-to-end joint training of the feature transformation network and the alignment prediction network of TAP.

SoftDTW is differentiable and can also be combined with different losses to train $h$. This can be achieved by replacing the TAP distance $d(h(\boldsymbol{X}^n), h(\boldsymbol{X}^{n'}); f)$ with the SoftDTW distance in Eq. (4),

---

[1] https://github.com/tobyperrett/few-shot-action-recognition

Table 2: Comparison of the training time per epoch (middle) and running time (bottom).

| Dataset | Action3D | SAD | ChaLearn |
|---------|----------|-----|----------|
| $l$ | 39.64 | 39.81 | 39.72 |
| SoftDTW | 23.28 | 508.39 | 544.30 |
| TAP | 35.78 | 876.75 | 901.46 |
| DTW | 0.9212 | 29.568 | 28.9270 |
| OT | 279.6414 | 794.9984 | 2614.677 |
| OPW | 0.8029 | 25.309 | 32.7774 |
| SoftDTW | 0.217 | 4.922 | 4.959 |
| TAP | **0.138** | **3.009** | **3.033** |

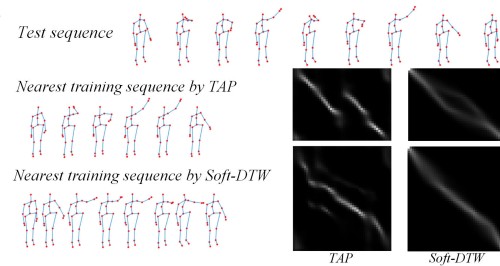

Figure 2: TAP (left) alignment and Soft-DTW (right) alignment.

Eq. (5), and Eq. (7), respectively. Comparisons between TAP and SoftDTW for supervised representation learning are also presented in Tab. 1. We employ the PyTorch implementation[2] of SoftDTW. The triplet loss is relatively straightforward, since the sequence distance does not undergo a non-linear function. SoftDTW outperforms TAP when the HM loss is used on some datasets. In the other two losses, exponential and logarithmic transformations are applied to the sequence distance, so the backpropagation through the alignment variable is more complicated. For SoftDTW, using a large $\alpha$ will lead to infinite initial losses, so we reduce $\alpha$ and $\beta$ to 2 and 1, respectively. When using the lifted structure loss and binomial deviance loss to optimize the transformation network $h$, the results of SoftDTW are inferior to those of TAP on most datasets. In addition, it is easy to fall into a trivial solution when incorporating SoftDTW into complex losses on some datasets. This may be because that gradients backpropagated through SoftDTW cannot be automatically calculated and such manually derived backward processes with respect to complex losses may be more unstable. On all datasets, TAP outperforms all other methods by a significant margin on MAP. In cases where the accuracy of TAP is not the best, it is also comparable with the best result. These results show that TAP can be better learned with different losses, the training of TAP is easier and more consistent, and TAP has better potential to be applied to more complex tasks. In addition, TAP can be generalized to tackle non-sequential data, e.g., determining the spatial or set-to-set correspondences; while SoftDTW is more restricted to strictly order-preserving sequence data.

## 4.4 EVALUATION ON COMPUTATIONAL TIMES AND ALIGNMENTS

We compare the average training time per epoch of SoftDTW and TAP on a single TITAN V GPU by using the triplet loss in the middle part of Table 2. Since the asymmetric TAP distance between any two sequences in a batch is calculated twice, the training time of TAP is longer than SoftDTW. Table 2 (bottom) compares the average running times for computing the distances between one test sequence and all training sequences with different distances including DTW, optimal transport (OT), OPW, SoftDTW, and TAP. $l$ in the top row indicates the average length of sequences in the dataset. TAP runs faster than OT, DTW, and OPW by at least an order of magnitude. The larger the dataset, the more significant the advantage. This is because TAP avoids complex optimizations required by other distances and can be accelerated conveniently by GPU. When both running on GPU, TAP is faster than SoftDTW. As shown in Appendix A.5, the advantage increases as $l$ increases.

Fig.2 shows visualizations on the MSR Action3D dataset. For the test sequence of "draw tick", the nearest training sequence by TAP comes from the same class, but the one by Soft-DTW comes from another class "horizontal arm wave". Actions contain different movement cycles and similar poses. Each alignment path of Soft-DTW is strictly order-preserving, causing some meaningless alignments. The alignments of TAP show periodicity while preserving the general orders.

In the alignment prediction network $f$, we implement $g$ by CNN to model the temporal constraints since multiple convolution kernels in different layers are suitable for capturing different local alignment patterns and variances of local motions. The local receptive field of CNN can be viewed as imposing a constraint on the local temporal warping. Ablation studies on the architecture and hyper-parameters of $g$ are presented in Appendix A.7. To evaluate the effect of $g(\boldsymbol{D}_s)$, we apply the trained

---

[2]https://github.com/Sleepwalking/pytorch-softdtw. In Fig. 2, we use the implementation (https://github.com/mblondel/soft-dtw) for calculating the SoftDTW alignment.

Table 5: Comparison of few-shot action recognition methods.

| Method | UCF101 | | HMDB51 | | SSv2 | |
|---|---|---|---|---|---|---|
| | 1-shot | 5-shot | 1-shot | 5-shot | 1-shot | 5-shot |
| ARN | 66.3±0.99 | 83.1±0.70 | 45.5±0.96 | 60.6±0.86 | - | - |
| CMN-J | - | - | - | - | 34.4 | 43.8 |
| OTAM | 79.9 | 88.9 | 54.5 | 66.1 | 42.8 | 52.3 |
| TRX($\Omega = 1$) | 79.0 | 95.9 | 52.1 | **76.1** | 38.8 | 60.6 |
| TRX($\Omega = 2, 3$) | 78.2 | **96.1** | 53.1 | 75.6 | 42.0 | **64.6** |
| TTAN | 80.9 | 93.2 | 57.1 | 74.0 | **46.3** | 60.4 |
| SoftDTW | 51.4±0.4 | 62.1±0.4 | 36.0±0.4 | 46.0±0.4 | 23.9±0.3 | 31.5±0.4 |
| TAP | 83.5±0.3 | 94.5±0.2 | 54.0±0.4 | 72.5±0.4 | 43.5±0.4 | 61.3±0.4 |
| SA-TAP | **83.9**±0.3 | 95.4±0.2 | **57.5**±0.4 | 74.2±0.4 | 45.2±0.4 | 63.0±0.4 |

TAP model on each test sequence and calculate the Frobenius norm ratio: $\frac{\|g(\boldsymbol{D}_s)\|_F}{\|\boldsymbol{D}\|_F}$. The average ratios of all test sequences with different losses are shown in Tab. 3. The ratio ranges from 2 to 21, showing that $g(\boldsymbol{D}_s)$ contributes more than $\boldsymbol{D}$.

Table 3: Frobenius Norm ratios.

| Method | Action3D | Activity3D |
|---|---|---|
| HM+TAP | 3.5539 | 10.2145 |
| lifted+TAP | 21.2767 | 15.6680 |
| Binomial+TAP | 2.1119 | 13.8124 |

Table 4: MSE between predicted alignments and constructed ground-truth alignments.

| Dataset | SoftDTW-s | TAP-s | SoftDTW | TAP |
|---|---|---|---|---|
| Action3D | 0.0165 | **0.0131** | 0.0163 | **0.0062** |
| Activity3D | 0.0040 | **0.0037** | 0.0037 | **0.0026** |

To quantitatively evaluate the alignments, we generate augmentations and corresponding ground-truth alignments for each sequence in the test set. Comparison on MSE between ground-truths and predicted alignments by the supervised learned TAP (TAP-s) and SoftDTW (SoftDTW-s) is shown in the left part of Tab. 4. $f$ in TAP can also be directly learned by minimizing MSE on the training set via augmentations. We directly apply TAP learned in this way and SoftDTW to align the original test sequences and their augmentations. Results are shown in the right part of Tab. 4. Details of the evaluation methods are presented in Appendix A.8. In both cases, TAP achieves lower MSEs.

## 4.5 EVALUATION ON FEW-SHOT ACTION RECOGNITION

We compare TAP and SA-TAP with state-of-the-art few shot action recognition methods including ARN Zhang et al. (2020), CMN-J (Zhu & Yang, 2018; 2020), OTAM (Cao et al., 2020), TRX (Perrett et al., 2021), and TTAN (Li et al., 2021). Results on 5-way 1-shot and 5-way 5-shot tasks are shown in Table 5, where the results of OTAM and TRX on UCF101 and HMDB51 are reported in (Li et al., 2021). OTAM employs a modified DTW as the distance. TAP generally outperforms OTAM and SoftDTW significantly. When only 1 support sample per class is available, TAP also achieves better results than TRX. TAP with much fewer parameters outperforms TTAN consisting of two stages on UCF101. TAP has tighter 95% confidence intervals and is more stable than SoftDTW and ARN. SA-TAP further improves TAP. It outperforms all other methods in 1-shot tasks on UCF-101 and HMDB-51. On SSv2, SA-TAP achieves comparable results to TTAN in 1-shot learning and comparable results to TRXs in 5-shot learning, while significantly outperforming other methods. These results validate the effectiveness of TAP and SA-TAP on few-shot action recognition.

## 5 CONCLUSION

In this paper, we present TAP, which is a learnable distance for sequences. TAP calculates the optimal alignment from the input sequences using a prediction network. We demonstrate the application of TAP in supervised representation learning for sequence data and few-shot action recognition. For supervised learning, TAP can be learned with various metric learning-based losses. Experimental results on four real-world datasets show that TAP achieves competitive performances with much faster inference speed. For few-shot learning, simply applying TAP in the metric-based strategy achieves comparable results with some state-of-the-art few-shot action recognition methods.

## ETHICS STATEMENT

This paper proposes a learnable sequence distance that employs a lightweight neural network to predict the optimal alignment between two sequences and its applications in supervised learning, few-shot learning, and self-supervised learning (in the appendix) for sequence data. The proposed distance can be applied to any sequence data to speed up the inference of optimal alignment and improve the discrimination ability. Because this work presents such a general distance measure and its learning methods under different conditions, we did not see any particular foreseeable negative ethics impacts.

## REPRODUCIBILITY STATEMENT

The architecture and hyper-parameters of the alignment prediction network in TAP are fixed and specified in Section 3.1. We specify all the training details on hyper-parameters and experimental settings in Section 4.2. Our code is available at `https://github.com/BingSu12/TAP`.

## ACKNOWLEDGMENTS

This work was supported in part by the National Natural Science Foundation of China No. 61976206 and No. 61832017, Beijing Outstanding Young Scientist Program NO. BJJWZYJH012019100020098, Beijing Academy of Artificial Intelligence (BAAI), China Unicom Innovation Ecological Cooperation Plan, the Fundamental Research Funds for the Central Universities, the Research Funds of Renmin University of China 21XNLG05, and Public Computing Cloud, Renmin University of China. This work was also supported in part by Intelligent Social Governance Platform, Major Innovation & Planning Interdisciplinary Platform for the "Double-First Class" Initiative, Renmin University of China, and Public Policy and Decision-making Research Lab of Renmin University of China.

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

## A  APPENDIX

### A.1  SELF-SUPERVISED LOSS FOR TAP

We show that the proposed TAP distance can also be learned in a self-supervised manner with unlabeled sequence samples. We learn the prediction network of TAP by minimizing the TAP dis-tances between sequences and their corresponding augmentations. Data of specific modalities such as images and videos can be augmented through low-level transformations, where self-supervised learning methods learn the feature embedding network by encouraging consistent features extracted from original data and their augmentations. Differently, we focus on learning the distance between sequences, where the element-wise features are already given no matter where and how the fea-tures are extracted, so low-level transformations are not applicable. To this end, we propose two augmentation methods for sequences: random blur and random merge.

**Random blur.** Random blur constructs a blur kernel $\boldsymbol{K}$ with a randomly selected size. The values in the kernel are also randomly selected from the range of $[0, 1]$ so that values in the kernel decrease from middle to both sides. Given a sequence $\boldsymbol{X} = [\boldsymbol{x}_1, \cdots, \boldsymbol{x}_{L_X}] \in \mathbb{R}^{d \times L_X}$, random blur generates an augmented sequence $\tilde{\boldsymbol{X}}$ by convolving $\boldsymbol{X}$ with $\boldsymbol{K}$. The stride along the 1st dimension is set to 1. The stride along the 2nd dimension is randomly selected from $1, 2, 3$. No padding is applied. Thus, each column element $\tilde{\boldsymbol{x}}_i$ of $\tilde{\boldsymbol{X}}$ is the weighted average of several successive elements of $\boldsymbol{X}$. The ground-truth alignment values between $\tilde{\boldsymbol{x}}_i$ and the $N_k$ elements are set as the values in $\boldsymbol{K}$, while those between $\tilde{\boldsymbol{x}}_i$ and all other elements of $\boldsymbol{X}$ are set to 0. In this way, we construct the ground-truth alignment matrix $\tilde{\boldsymbol{T}}$ between $\boldsymbol{X}$ and $\tilde{\boldsymbol{X}}$.

**Random merge.** For $\boldsymbol{X}$ with a length of $L_X$, random merge first randomly selects an integer $\tilde{L}$ from $[0.5L_X, 0.8L_X]$. $\boldsymbol{X}$ is then randomly segmented into $\tilde{L}$ parts, where each part has at least one element. For the $k$-th part with $\tilde{L}_k$ elements, $\tilde{L}_k$ values are randomly sampled from a standard Gaussian distribution and then normalized by softmax to form $\tilde{L}_k$ weights. The $\tilde{L}_k$ elements are averaged with the weights to form a new element $\tilde{\boldsymbol{x}}_k$. The augmented sequence consists of all the generated elements for all parts sequentially: $\tilde{\boldsymbol{X}} = [\tilde{\boldsymbol{x}}_1, \cdots, \tilde{\boldsymbol{x}}_{\tilde{L}_k}]$. The ground-truth alignment values between $\tilde{\boldsymbol{x}}_k$ and the $\tilde{L}_k$ elements in the $k$-th part of $\boldsymbol{X}$ are set as the $\tilde{L}_k$ corresponding weights, while those between $\tilde{\boldsymbol{x}}_k$ and elements in other parts are set to 0. In this way, the ground-truth alignment matrix $\tilde{\boldsymbol{T}}$ between $\boldsymbol{X}$ and $\tilde{\boldsymbol{X}}$ is constructed.

**Self-supervised learning of TAP.** Given $N$ unlabeled training sequences $\boldsymbol{X}^n, n = 1, \cdots, N$, the goal is to learn the TAP distance by learning the prediction network. For each $\boldsymbol{X}^n$, we first ran-domly select one of the two methods to generate its augmented sequence $\tilde{\boldsymbol{X}}^n$ and the corresponding

ground-truth alignment $\tilde{\boldsymbol{T}}^n$. We then calculate the TAP distance $d(\boldsymbol{X}^n, \tilde{\boldsymbol{X}}^n; f)$ and the corresponding non-normalized alignment $\boldsymbol{A}^n$ in Eq. (3) between $\boldsymbol{X}^n$ and $\tilde{\boldsymbol{X}}^n$. Since TAP is asymmetric, we also calculate the dual TAP distance $d(\tilde{\boldsymbol{X}}^n, \boldsymbol{X}^n; f)$ and dual alignment $\boldsymbol{A}_d^n$ between $\tilde{\boldsymbol{X}}^n$ and $\boldsymbol{X}^n$. We minimize the TAP and its dual distances. We also minimize the Mean Squared Errors (MSEs) between the predicted dual alignments and the ground-truth alignments, which indirectly forces the alignment and the dual alignment to be consistent. The objective for a batch of $B$ sequences is:

$$\min_f \frac{1}{B} \sum_{n=1}^{B} (d(\boldsymbol{X}^n, \tilde{\boldsymbol{X}}^n; f) + d(\tilde{\boldsymbol{X}}^n, \boldsymbol{X}^n; f) + MSE(\boldsymbol{A}^n, \tilde{\boldsymbol{T}}^n) + MSE(\boldsymbol{A}_d^n, \tilde{\boldsymbol{T}}^{nT})). \quad (10)$$

Parameters of the alignment prediction network $f$ in TAP are learned by back-propagation. Since $\boldsymbol{D}$ is fixed with given sequences and $\boldsymbol{T}$ is obtained after softmax and $L_1$ normalization, the TAP distance between two sequences is bounded. Therefore, $f$ has no trivial collapse solution, so negative sequence samples are naturally not required. Once $f$ is learned, TAP can be applied to measure the difference between two sequences by feedforward calculations. In Section 3.2, the self-supervised loss 10 is added to all supervised losses to improve the stability of the learned TAP.

Table 6: Results on the MSR Action3D dataset.

| Distance | MAP | 1-NN | 5-NN | 30-NN |
|---|---|---|---|---|
| DTW | **0.590** | 81.32 | **80.95** | 72.53 |
| lDTW | 0.567 | 82.78 | 79.12 | 64.84 |
| nDTW | 0.565 | 79.85 | 76.92 | 67.40 |
| OT | 0.544 | 78.02 | 75.09 | 59.34 |
| Sinkhorn | 0.546 | 78.02 | 74.73 | 60.44 |
| TCOT | 0.578 | 80.59 | 79.49 | 67.03 |
| OPW | 0.587 | **84.25** | 80.22 | 67.03 |
| SoftDTW | 0.589 | 81.32 | **80.95** | **72.89** |
| SoftDTW* | 0.076 | 4.03 | 5.50 | 5.13 |
| TAP | 0.557 | 79.85 | 78.02 | 67.77 |

Table 7: Results on MSR Activity3D dataset.

| Distance | MAP | 1-NN | 5-NN | 30-NN |
|---|---|---|---|---|
| DTW | 0.338 | 58.75 | 49.38 | 31.87 |
| lDTW | 0.288 | 50.00 | 50.00 | 30.00 |
| nDTW | 0.306 | 55.63 | 52.50 | 30.00 |
| OT | 0.308 | 55.63 | 50.62 | 25.00 |
| Sinkhorn | 0.307 | 54.37 | 50.62 | 25.62 |
| TCOT | 0.321 | 59.38 | 51.88 | 28.13 |
| OPW | 0.346 | 58.13 | 53.75 | 30.63 |
| SoftDTW | 0.339 | 60.00 | 50.00 | 33.75 |
| SoftDTW* | 0.108 | 6.25 | 6.25 | 6.25 |
| TAP | **0.362** | **61.25** | **55.63** | **36.25** |

**Comparison with alignment-based distances.** We compare the self-supervised learned TAP with 8 alignment-based sequence distances in the retrieval and classification tasks, including DTW, length-normalized DTW (lDTW), matching step-normalized DTW (nDTW), optimal transport (OT) (Kolouri et al., 2017), Sinkhorn (Cuturi, 2013), TCOT (Su & Hua, 2019), OPW (Su & Hua, 2019), and SoftDTW (Cuturi & Blondel, 2017). We follow the setting in (Su & Hua, 2019) and directly compare with the results reported in Su & Hua (2019). The comparisons on the MSR Action3D and MSR Activity3D datasets are shown in Table 6 and Table 7, respectively. We observe that TAP generally outperforms other distances on the MSR Activity3D dataset, but performs worse on the MSR Action3D dataset.

SoftDTW can be used as a differentiable loss on top of a single-layer embedding. SoftDTW* trains the additional linear layer in a self-supervised manner by minimizing the SoftDTW distances between the original training sequences and their augmented ones, but the effect is equivalent to random classification. Without negative sample sampling, SoftDTW* falls into the trivial solution of all zeros. In self-supervised learning, TAP naturally avoids the trivial solution since it calculates $\boldsymbol{D}$ from the original sequences and performs Softmax on $\boldsymbol{T}$.

### A.2 PROBLEM OF SUPERVISED REPRESENTATION LEARNING WITH ALIGNMENT-BASED DISTANCES

For supervised learning, the objective is often to make the sequence distance better separate sequences from different classes. However, such objectives are difficult to optimize for conventional alignment-based distances because they solve another optimization problem to obtain the alignments. For example, if we apply the triplet loss to the OPW distance, the objective becomes:

$$\min_h \sum_{n=1}^{B} (\sum_{n':y^{n'}=y^n} \max(\langle \boldsymbol{T}_{nn'}^*, \boldsymbol{D}(h(\boldsymbol{X}^n), h(\boldsymbol{X}^{n'})) \rangle - d_s^n + m, 0)$$

$$+ \sum_{n':y^{n'}\neq y^n} \max(d_l^n + m - \langle \boldsymbol{T}_{nn'}^*, \boldsymbol{D}(h(\boldsymbol{X}^n), h(\boldsymbol{X}^{n'})) \rangle, 0)) \quad . \tag{11}$$

$$s.t. \ \boldsymbol{T}_{nn'}^* = \arg\min_{\boldsymbol{T}\in\Phi} \langle \boldsymbol{T}, \boldsymbol{D}(h(\boldsymbol{X}^n), h(\boldsymbol{X}^{n'})) \rangle + \mathcal{R}(\boldsymbol{T})$$

The gradients of this objective with respect to the parameters of $h$ rely on $\boldsymbol{T}_{nn'}^*$, which is determined by another optimization problem with additional constraints on $\boldsymbol{T}$ and this optimization problem depends on $h$ in turn. Therefore, the gradients cannot be explicitly formulated and directly obtained. The overall optimization problem thus cannot be optimized via back propagation in an end-to-end manner. An alternative solution is to alternatively update $\boldsymbol{T}^*$ and $h$ by fixing the other in an iterative way, which is time-consuming and there is no theoretical guarantee of convergence. Existing methods (Mei et al., 2014; Su & Wu, 2020b) optimize a proxy or approximate objective through complex iterative solutions.

### A.3 SELF-ATTENTION MODULE IN SA-TAP

In few-shot action recognition, TAP can be combined with the self-attention mechanism. Especially, the proposed TAP focuses on aligning different sequences, but cannot fully utilize the temporal dependencies among frames in the same sequence. This can be alleviated by applying a single-head self-attention (SA) module to each sequence before performing alignment by TAP. Specifically, after obtaining the feature sequence $\boldsymbol{X}$ for a video, we first obtain $\boldsymbol{X}_p = \boldsymbol{X} + P_e(\boldsymbol{X})$, where $P_e(\boldsymbol{X})$ is the sequence of position embeddings for elements in $\boldsymbol{X}$. We then employ the self-attention module to transform $\boldsymbol{X}_p$ into the following sequence:

$$\boldsymbol{X}_a = Softmax(\frac{\boldsymbol{X}_p^T \boldsymbol{W}_q (\boldsymbol{X}_p^T \boldsymbol{W}_k)^T}{\sqrt{d}}) \boldsymbol{X}_p^T \boldsymbol{W}_v, \tag{12}$$

where $\boldsymbol{W}_q$, $\boldsymbol{W}_k$, and $\boldsymbol{W}_v$ are three linear projections with the size of $d \times d$.

The sequence $\boldsymbol{X}$ is finally transformed into:

$$\boldsymbol{X} = LN(\boldsymbol{X}_a^T + \boldsymbol{X}_p) \tag{13}$$

Where $LN$ is the layer normalization. Both the support and query sequences are enhanced before calculating their TAP distance in Eq. (8). We denote this method by SA-TAP.

### A.4 COMPLEXITY AND MODEL SIZE

The complexities for calculating $\boldsymbol{D}$, performing convolutions with a few layers and small kernels, calculating $\boldsymbol{S}$, and performing Softmax and $L_1$ normalization are $O(L_X L_Y d)$, $O(L_X L_Y d)$, $O(L_X L_Y)$, and $O(L_X L_Y)$, respectively, where the number of layers, the size of kernels, and the number of kernels are omitted since they are small. The overall complexity of TAP is $O(L_X L_Y d)$.

TAP only contains three convolution layers and hence the model size is quite small. For supervised learning, the feature transformation network is the same for all methods. DTW, OPW, and Soft-DTW are parameter-free but require solving an optimization problem for inferring the optimal alignment, so they introduce no additional parameters. In the three layers of TAP, we fix the kernel sizes to 5, 5, and 3, respectively, and we fix the number of filers to 30, 30, and 1, respectively. Therefore, TAP only introduces $30 \times 5 \times 5 \times 2 + 30 \times 5 \times 5 \times 30 + 1 \times 3 \times 3 \times 30 = 0.0231M$ parameters.

For few-shot action recognition, the feature extraction backbone is fixed to ResNet-50 for all methods. OTAM employs a variant of DTW as the distance measure and hence does not introduce additional parameters. TRX contains two $\Omega \times d \times d_k$ projections and one $\Omega \times d \times d_v$ projection, where $\Omega$ is the length of a tuple, $d = 2048$ is the dimension of frame-wide features and $d_k = d_v$ is set to 1152. Therefore, TRX introduces $3 \times 2 \times 2048 \times 1152 \times \Omega$ parameters, $\Omega$ is often set to 2 for using frame pairs and 3 for frame tuples, so the number of parameters ranges from 27M to 40.5M. Moreover, TRX requires an exhaustive traversal of pairs and triplets of frames with high complexity. In TTAN, the location network in the temporal transform module takes a feature sequence as input

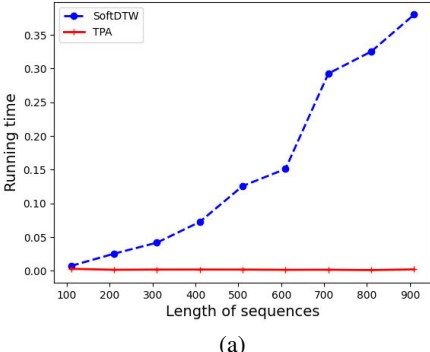 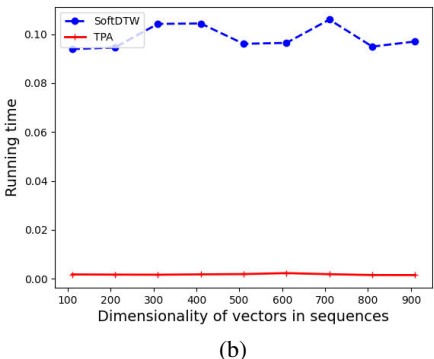

(a)   (b)

Figure 3: (a) Running times of SoftDTW and TAP as functions of the length of sequences. (b) Running times of SoftDTW and TAP as functions of the dimensionality of features in sequences.

Table 8: Comparison of TAP with/without data augmentation for supervised learning.

| Method | Action3D | | Activity3D | |
|---|---|---|---|---|
| | MAP | 1-NN | MAP | 1-NN |
| HM+TAP w/o DA | 0.839 | 88.28 | 0.690 | 71.25 |
| HM+TAP | 0.798 | 87.18 | 0.701 | 74.38 |
| Lifted+TAP w/o DA | 0.820 | 86.81 | 0.723 | 70.00 |
| Lifted+TAP | 0.772 | 82.05 | 0.728 | 73.75 |
| Binomial+TAP w/o DA | 0.802 | 86.45 | 0.689 | 71.88 |
| Binomial+TAP | 0.826 | 87.91 | 0.678 | 73.13 |

and outputs two indexes with several learnable layers, but the detailed network structure is not specified; the temporal attentive module contains 3 $d \times d$ projections and has 12M parameters; the late fusion module introduces another 3 $d \times d$ projections with 12M parameters. In contrast, TAP-CNN also only introduces $64 \times 5 \times 5 \times 2 + 64 \times 5 \times 5 \times 64 + 1 \times 3 \times 3 \times 64 = 0.1013M$ parameters since the kernel sizes are enlarged to 64, 64, and 1, respectively. For SA-TAP, the self-attention layer contains 3 $d \times d$ projections, Therefore, SA-TAP introduces $3d^2 + 0.1013M = 12.1013M$ parameters. The model sizes of both TAP and SA-TAP are smaller than TRX and TTAN.

### A.5 COMPARISON OF TAP AND SOFTDTW ON RUNNING TIMES

We study the scalability of the proposed TAP in comparison with SoftDTW as the length and dimensionality increase. We randomly generate two sequences of length $L$, both consists of $d$-dimensional features. For different $L$ and $d$, we compute the SoftDTW distance and the TAP distance with random parameters between the two sequences, respectively, and compare the corresponding running times. We first fix $d$ to 500, and increase $L$ from 100 to 1000 with an interval of 100. The running times of SoftDTW and TAP as functions of $L$ are compared in Fig. 3(a). The running time of SoftDTW grows rapidly as the length increases due to the dynamic programming and backtracking processes. In contrast, since only the forward matrix operations are required, the running time of TAP does not change much as the length increases. The longer the sequences, the more obvious the speed advantage of the proposed TAP.

We then fix $L$ to 500, and increase $d$ from 100 to 1000 with an interval of 100. The running times of SoftDTW and TAP as functions of $d$ are compared in Fig. 3(b). We observe that the proposed TAP is consistently about 10 times faster than SoftDTW. These results demonstrate that TAP can better scale to long and high-dimensional sequences.

Table 9: Comparison of using transformer and CNN.

| Method | Action3D | | Activity3D | |
|---|---|---|---|---|
| | MAP | 1-NN | MAP | 1-NN |
| HM+TAP-Trans | 0.751 | 81.32 | 0.691 | 72.50 |
| HM+TAP(CNN) | 0.798 | 87.18 | 0.701 | 74.38 |
| Lifted+TAP-Trans | 0.740 | 77.29 | 0.697 | 69.38 |
| Lifted+TAP(CNN) | 0.772 | 82.05 | 0.728 | 73.75 |
| Binomial+TAP-Trans | 0.746 | 80.22 | 0.705 | 71.25 |
| Binomial+TAP(CNN) | 0.826 | 87.91 | 0.678 | 73.13 |

### A.6 EFFECT OF DATA AUGMENTATION ON SUPERVISED LEARNING

We apply the data augmentations, i.e., random blur and random merge, in supervised learning only for obtaining more stable results. For example, in some cases, all sequences in a small batch satisfy the constraints in Eq. (4) and Eq. (6), where data augmentation can enlarge the within-class distance. Comparison results of TAP with and without data augmentations on the MSR Action3D and MSR Activity3D datasets are shown in Tab. 8. In some cases, the data augmentation only brings minor improvements, while in other cases, the data augmentation even causes performance degradation, which may be because augmented sequences obtained by such temporal blur and merge do not reflect the real intra-class variances. Generally, in most cases, since the augmented sequence often has a small distance from the original sequence, it will not change the selection of hard negatives and hard positives in Eq. (6). Without the proposed data augmentations, TAP still outperforms all other methods where it performs best in Tab. 1, and all conclusions still hold.

### A.7 ABLATION ON THE ARCHITECTURE OF THE ALIGNMENT PREDICTION NETWORK

The intuition behind using CNN as $g$ in the alignment prediction network $f$ is that CNN can implicitly model the temporal constraints and capture local alignment patterns. Temporal constraints for performing temporal alignment are often reflected as follows: the probability of two frames being aligned is not only related to the distance between the two elements and the difference between their relative positions, but also depends on the distances and alignment probabilities between their adjacent frames. Convolution kernels aim at capturing such local information. There may exist different local alignment patterns at different levels. Multiple convolution kernels in different layers are suitable for capturing these different local patterns and variances of local motions.

A transformer-like cross-attention-based architecture can also be used to predict temporal alignment. Specifically, for aligning $\boldsymbol{Y}$ to $\boldsymbol{X}$, we use a linear projection $\boldsymbol{W}_q \in \mathbb{R}^{d' \times d}$ to transform all elements $\boldsymbol{x}_i$ of $\boldsymbol{X}$ to $\hat{\boldsymbol{x}}_i = \boldsymbol{W}_q \boldsymbol{x}_i$, resulting in a transformed sequence $\hat{\boldsymbol{X}} = [\hat{\boldsymbol{x}}_i, i = 1, \cdots, L_X] = \boldsymbol{W}_q \boldsymbol{X}$, where $d' = 30$ is the transformed dimensionality. Similarly, $\boldsymbol{Y}$ is transformed into $\hat{\boldsymbol{Y}} = [\hat{\boldsymbol{y}}_j, j = 1, \cdots, L_Y] = \boldsymbol{W}_k \boldsymbol{Y}$ by another projection $\boldsymbol{W}_k$. We use each $\hat{\boldsymbol{x}}_i$ as the query, use $\hat{\boldsymbol{y}}_j, j = 1, \cdots, L_Y$ as keys, and calculate all pairwise Euclidean distances between $\hat{\boldsymbol{x}}_i, i = 1, \cdots, L_X$ and $\hat{\boldsymbol{y}}_j, j = 1, \cdots, L_Y$ into a matrix $\boldsymbol{D}_s := [e(\hat{\boldsymbol{x}}_i, \hat{\boldsymbol{y}}_j)/\sqrt{d'}]_{ij} \in \mathbb{R}^{L_X \times L_Y}$. The similarity matrix $\boldsymbol{S}$ is calculated as: $\boldsymbol{S} = -(\boldsymbol{D}_s + \lambda \boldsymbol{D}_t)$, where the hyper-parameter $\lambda$ is set to 50. Similarly with TAP, we perform Softmax and $L_1$ normalization in Eq.(3) to $S$ to obtain the final alignment $T$. We denote this method by TAP-trans. Comparisons of TAP and TAP-trans are shown in Tab. 9. We observe that CNN outperforms the Transformer-like attention-based network. This may indicate that CNN indeed can capture the local alignment patterns by integrating the local contexts.

The local receptive field of CNN can be viewed as imposing a constraint on the temporal deviation: only local warping within the receptive field is considered since extremely unbalanced warping is not preferred. Increasing the kernel sizes or using more convolution layers gains larger receptive fields, hence larger warping patterns are allowed and long-range dependencies can be utilized. We have added comparisons with two variants, namely TAP (l) and TAP (d), to evaluate the effects of using larger kernels and using more convolution layers, respectively. In TAP (l), the kernel size in the three convolution layers is increased to 7, 7, and 5 with paddings 3, 3, and 2 to keep the spatial

Table 10: Effects of using larger kernels and more convolutional layers in the alignment prediction network of TAP.

| Method | Action3D | | Activity3D | |
|---|---|---|---|---|
| | MAP | 1-NN | MAP | 1-NN |
| HM+TAP | 0.798 | 87.18 | 0.701 | 74.38 |
| HM+TAP(l) | 0.834 | 90.48 | 0.671 | 73.13 |
| HM+TAP(d) | 0.846 | 91.58 | 0.693 | 70.63 |
| Lifted+TAP | 0.772 | 82.05 | 0.728 | 73.75 |
| Lifted+TAP(l) | 0.805 | 85.35 | 0.743 | 73.75 |
| Lifted+TAP(d) | 0.779 | 83.52 | 0.743 | 71.25 |
| Binomial+TAP | 0.826 | 87.91 | 0.678 | 73.13 |
| Binomial+TAP(l) | 0.838 | 85.348 | 0.701 | 72.50 |
| Binomial+TAP(d) | 0.856 | 89.377 | 0.678 | 74.38 |

size, respectively. In TAP (d), another convolution layer is inserted after the second layer, where the kernel size, padding, and the number of kernels are set to 5, 2, and 30, respectively.

Comparison results are shown in Tab. 10. Generally, using larger convolution kernels or adding more convolution layers to relax the restriction on warping and model more levels of alignment patterns can further increase the performance of TAP. We use the CNN architecture consisting of 3 convolution layers with small kernels to impose strong inductive bias and keep the alignment prediction network lightweight.

## A.8 EVALUATION METHODS ON MSE BETWEEN PREDICTED AND GROUND-TRUTH ALIGNMENTS.

For each test sequence, we randomly generate 5 augmentations with the proposed augmentation methods (random blur and random merge) and approximately construct the ground-truth alignment between the original sequence and its augmentations, respectively. To evaluate the predicted alignments by supervised trained TAP (TAP-s), we first apply the trained transformation network $h$ to transform the sequence and each of its augmentations, and then predict their un-normalized alignments by the trained alignment prediction network $f$. For supervised learning with SoftDTW (SoftDTW-s), after transforming the sequences with the trained $h$, the alignments are directly inferred by SoftDTW. The MSE between the predicted alignments and the ground-truth alignments of all test-augmentation sequence pairs is calculated as the performance measure. Comparisons of TAP-s and SoftDTW-s by using the triplet loss are shown in the left part of Tab. 4.

We can also directly train TAP without jointly learning the transformation network $h$ by constraining the predicted alignments. We generate augmentations for each training sequence and construct the corresponding ground-truth alignment. The prediction network $f$ is trained by minimizing the MSE loss between the predicted un-normalized alignments by TAP and the constructed ground-truth alignments. In the testing stage, we directly apply SoftDTW or the trained TAP to predict the alignments between test sequences and their corresponding augmentations. Comparisons on the MSE between the ground-truth alignments and the alignments predicted by SoftDTW and TAP on the test set are shown in the right part of Tab. 4. We observe that in both cases, TAP achieves lower MSEs, which may indicate that TAP can indeed learn alignment patterns from the training sequences to better approximate the ground-truth alignments.

## A.9 EVALUATING THE LEARNED REPRESENTATIONS WITH A LINEAR CLASSIFIER

In supervised representation learning, we jointly learn the frame-wise feature extraction network $h$ and the alignment prediction network $f$. We further use a linear classifier to evaluate the learned representations. For each sequence, we freeze the backbone to extract frame-wise features and then encode them into a vector representation by mean pooling. We then train a linear classifier on these vector representations. We use the Adam optimizer, where the hyper-parameters of the optimizer remain the same as in supervised representation learning. The number of epochs is fixed to 3000. Comparisons of TAP and SoftDTW with the linear classifier are shown in Tab. 11.

Table 11: Comparison of SoftDTW and TAP with the linear classifier.

| Method | Action3D | | Activity3D | |
|---|---|---|---|---|
| | Top-1 | Top-5 | Top-1 | Top-5 |
| HM+SoftDTW | 56.41 | 86.81 | 62.50 | 90.63 |
| HM+TAP | 67.03 | 90.84 | 63.75 | 91.25 |
| Lifted+SoftDTW | 5.50 | 24.54 | 6.25 | 31.25 |
| Lifted+TAP | 51.65 | 82.78 | 64.38 | 86.25 |
| Binomial+SoftDTW | 53.48 | 85.35 | 6.25 | 31.25 |
| Binomial+TAP | 69.23 | 90.11 | 65.63 | 91.25 |

We observe that TAP always outperforms SoftDTW with the linear classifier no matter what supervised representation learning loss is adopted and whether SoftDTW falls into the trivial solution. This shows that the representations learned by TAP are more discriminative. The results obtained by the linear classifier are worse than the 1-NN classifier, which indicates that the mean pooling loses useful temporal information and TAP can take advantage of such information.

