# OpenReview forum: "Temporal Alignment Prediction for Supervised Representation Learning and Few-Shot Sequence Classification"
_ICLR.cc/2022/Conference — ICLR 2022 Poster_

### Official Review · Reviewer_UnWN · 2021-10-25

**Correctness:** 3
**Technical Novelty And Significance:** 3
**Empirical Novelty And Significance:** 2
**Recommendation:** 6
**Confidence:** 3

**Main Review:**

This paper is interesting because it attempts to solve the alignment problem directly by using CNN. The problem is formulated as in Eqs. (1) and (2), and what we have to do is to find the optimal alignment matrix T*. This is usually obtained by solving an optimization problem, which prevents using backprop in an end-to-end manner.

The alignment matrix T is a rectangular matrix, so the proposed method directly estimates this matrix with CNN, whose inputs are matrix D (pairwise distance matrix of sequences X and Y) and distance matrix Dt of relative positions (larger when indices are different). These are stacked to Ds and fed to a small network g to generate g(Ds). With a residual connection, S=D+g(Ds) is further normalized to obtain an estimate of T*.

This is a quite simple idea, but sounds reasonable if we have a large amount of data. There could be a variety of potential applications.


There are some concerns.
First, the CNN g looks to have a relatively small receptive field compared to the length of sequences. In p.4, the CNN g has three layers with kernel sizes of 5,5,3, which indicates that the size of the receptive field is about 10 (because the spatial size is kept). However the datasets used in the experiments have much longer sequence lengths, as illustrated in Figure 2. This is strange when we consider the good performance of the proposed model.
(note: this applies only for the supervised learning, because few-shot learning uses only 8 frames)

Second, relating to the above, the CNN g might not work as expected. The residual connection D + g(Ds) works as shown in experiments, and it is not clear how much the residual g(Ds) contributes the results. If g(Ds) is small compared to D, the performance is mainly due to D, a simple distance matrix.
Since the backbone network h is jointly trained, the good performances mainly come from the backbone, not the alignment prediction network f.



minor issues:
It is interesting to see how the alignment prediction network correctly predicts alignments. It is possible to compare the prediction and the ground truth by solving optimization. Of course the correct prediction might not lead to higher performances, but checking the "correctness" of the prediction may show show some insights how the network behaves.


**Summary Of The Paper:**

One-shot sequence alignment with CNN.
Comparisons of sequence data are important for many tasks including action recognition and retrieval. Previous approaches include generating fixed-size feature vectors (e.g., RNN) or temporally aligning two sequences (e.g., DTW).
Instead, the proposed method directly predict the alignment matrix T between two sequences X and Y with CNN.
This is differentiable, and hence can be applied to many tasks such as supervised sequence classification/retrieval, and few-shot classification.
Experiments uses a variety of sequence classification tasks, such as skeleton-based action recognition and spoken digit audio classification, and show a good performance of the proposed method.

**Summary Of The Review:**

The concept is interesting, and potential applications are many.
However the proposed concept has not been well studied.

---

### Official Review · Reviewer_1HmZ · 2021-11-01

**Correctness:** 3
**Technical Novelty And Significance:** 3
**Empirical Novelty And Significance:** 3
**Recommendation:** 8
**Confidence:** 4

**Main Review:**

The paper is well motivated and focuses on an important problem. The method itself is simple and makes use of previous contributions (loss functions, alignment formulation and models), however its overall combined architecture and approach is novel and can provide inspiration for future work.

Nevertheless, I have some questions regarding the very nature of the problem. In fact, the model is trained to learn an optimal alignment between temporal sequences, yet there is no specific evaluation of the model for this very task. The authors provide a good number of experiments for two different settings, however there is no study showing how well the model is able to predict the optimal alignment between two sequences. While this may not be a major concern per se, it seems a little confusing that the method is trained for sequence alignment but it is not evaluated for that. In fact, the alignment here is proposed as a “proxy distance” to be used for other tasks. There is nothing particularly wrong with that, but again this missing study raises some confusion. I appreciate the examples in Figure 2 showing some alignments, but I reckon a quantitative analysis is needed.

The choice of CNN as the main model is a little questionable in my opinion. By design, CNNs work on fixed-size input. In fact, when the method is evaluated on few-show action recognition, a fixed number of frames is sampled from all sequences. For supervised representation learning it is not clear whether the input distance matrices are padded to have consistent input shapes or whether a fixed number of frames were sampled here too. While using fixed representation/samples for sequence alignment is not uncommon, the main reason why a CNN was preferred was its simplicity and interpretability. The latter is often mentioned in the paper, yet there is no analysis of the presented experiments with regards to interpretability. Considering this, the argument to favour CNNs becomes weaker and I wonder how other architectures better suited for sequence modelling (e.g. RNNs or Transformers) would work within the framework.

Regarding supervised representation learning, it is not clear why a simple 1-NN classifier was used. In fact, the typical approach in this case would be to train the backbone with metric learning, freeze then the backbone and train a linear classifier on the frozen distribution.

Finally, regarding running time, the authors compare inference speed for several methods, showing that theirs is fastest. I believe that reporting training time would be fair too or at very least interesting: since most of the methods compared against does not require training, it would be good to know what’s the training overhead of this model.

**Summary Of The Paper:**

This paper focuses on temporal sequence alignment, i.e. the task of finding an optimal alignment between sequences of different lengths. This task has been addressed by various traditional methods (e.g. DTW) that involve dynamic programming or other optimisations techniques that cannot be easily embedded in end-to-end learning frameworks.

This work proposes a method where the optimal temporal alignment is learnt by a CNN. The input is formed concatenating two matrices measuring element-wise distances between two sequences A and B. The network is trained to output a matrix whose (i, j)-th element indicates the likelihood of Ai being aligned to Bj.

The framework is evaluated with two tasks: supervised representation learning and few-shot action recognition. Tested on several datasets, the method is able to attain competitive performance and fast inference speed.

**Summary Of The Review:**

The paper is well written, addresses an interesting problem and proposes a novel approach for end-to-end temporal alignment. However the evaluation of the method, while comprehensive in some ways, has some holes. This may be a concern since the very nature of the method, temporal alignment, is not evaluated.

# Update after rebuttal

I believe the authors did a good job with their rebuttal. All my concerns have been well addressed and the paper is know more polished. I believe the quality of this work is good enough for publication.

---

### Official Review · Reviewer_tA1W · 2021-11-03

**Correctness:** 3
**Technical Novelty And Significance:** 2
**Empirical Novelty And Significance:** 3
**Recommendation:** 6
**Confidence:** 3

**Main Review:**

Strengths:

1. Using learnable sequence distance removes ordering constraints in DTW-based alignments, and this may be a better setting for actions that do not enforce strict ordering of motions.

2. As shown in Table 2, the inference speed of TAP is better than previous approaches.

3. The technical part is well written and easy to understand.


Weaknesses:

1. No significant improvement on few-shot action recognition performance. As shown in Table 3, TAP only outperforms prior work on UCF101 1-shot accuracy, while falls behind on other metrics. Moreover, many exisiting methods (e.g. OTAM, ARN, CMN, TTAN) benchmark on much larger action recognition datasets including Kinetics and Something-something, while there is no reported experimental results of TAP on these datasets. As those prior work of sequence distances focuses on few-shot action recognition, it is hard to convince me that TAP is superior than the prior work.

2. No ablation on the data augmentation. The authors also propose two augmentation methods, random blur and random range, and have used them in their experiments. However I don't see ablation study examining the contributions of the data augmentation. I can't tell if the performance improvement is mainly from TAP or the data augmentation.

3. Missing reference: Zhu and Yang, Compound Memory Networks for Few-shot Video Classification. ECCV 2018


Questions:

1. As shown in Table 1, TAP does better in terms of MAP, but not as good as the prior work in terms of 1-NN. Could you explain why TAP typically performs better on one metric over another?

2. If there's no data augmentation used, what is TAP's performance on both tasks?

3. TAP has an extra neural net for predicting alignment. What's the overall model size of TAP compared to prior approaches?

**Summary Of The Paper:**

This paper proposes a sequence distance metric named temporal alignment prediction (TAP) which leverages a CNN to predict alignment between two sequences, in contrast to other non-learnable alignment algorithms in prior work. The authors have benchmarked the TAP on two video tasks of supervised video representation learning and few-shot action recognition.

**Summary Of The Review:**

Although the proposed way of computing distances between sequences is learnable, flexible, and light-weighted, the experiments on few-shot action recognition could not support the supremacy of TAP over existing methods. The paper also lacks necessary ablation study on the data augmentation used in TAP experiments. Overall, I think the quality of the current draft is marginally lower than the acceptance threshold.

====== Post-rebuttal ======
The authors have addressed my concerns and questions in the rebuttal. I'm raising my rating to 6.

---

### Decision · Program_Chairs · 2022-01-20

**Decision:**

Accept (Poster)

**Comment:**

This paper has been evaluated by three reviewers with 2 borderlines leaning towards the accept, and with 1 accept. The reviewers have noted that the idea of alignment is not particularly novel per se. Nonetheless, they found some merit in the use of a network learning the alignment and they liked experiments.

AC has however some concerns about this work. Firstly, it is not clear why Lifted+SoftDTW and Binomial+SoftDTW completely fail in Table 1, and in Table 5, SoftDTW is worse by 30% than TAP. Is soft-DTW set up properly in these experiments (the softmax temperature, the base distance used, the maximum number of steps away from the main path etc.)?

AC is also not convinced about the principled nature of the proposed alignment. Eq. 3 and the residual design above seem more as heuristics than a principled OT transport as Eq. 1 and 2 set out to suggest. With concatenation of distances between sequence features and positional encoding, the proposed alignment seems more similar to attention and transformers than OT.